# Advancement in Diagnostic Imaging of Thymic Tumors

**DOI:** 10.3390/cancers13143599

**Published:** 2021-07-18

**Authors:** Francesco Gentili, Ilaria Monteleone, Francesco Giuseppe Mazzei, Luca Luzzi, Davide Del Roscio, Susanna Guerrini, Luca Volterrani, Maria Antonietta Mazzei

**Affiliations:** 1Unit of Diagnostic Imaging, Department of Radiological Sciences, Azienda Ospedaliero-Universitaria Senese, 53100 Siena, Italy; francescogmazzei@gmail.com (F.G.M.); guerrinisus@gmail.com (S.G.); 2Unit of Diagnostic Imaging, Department of Medical, Surgical and Neuro Sciences and of Radiological Sciences, University of Siena, Azienda Ospedaliero-Universitaria Senese, 53100 Siena, Italy; hilarymont@gmail.com (I.M.); davide.delroscio90@gmail.com (D.D.R.); lucavolterrani@gmail.com (L.V.); mamazzei@gmail.com (M.A.M.); 3Thoracic Surgery Unit, Department of Medical, Surgical and Neuro Sciences, University of Siena, Azienda Ospedaliero-Universitaria Senese, 53100 Siena, Italy; l.luzzi@ao-siena.toscana.it

**Keywords:** thymoma imaging, CT, thymoma diagnosis

## Abstract

**Simple Summary:**

Diagnostic imaging is pivotal for the diagnosis and staging of thymic tumors. It is important to distinguish thymoma and other tumor histotypes amenable to surgery from lymphoma. Furthermore, in cases of thymoma, it is necessary to differentiate between early and advanced disease before surgery since patients with locally advanced tumors require neoadjuvant chemotherapy for improving survival. This review aims to provide to radiologists a full spectrum of findings of thymic neoplasms using traditional and innovative imaging modalities.

**Abstract:**

Thymic tumors are rare neoplasms even if they are the most common primary neoplasm of the anterior mediastinum. In the era of advanced imaging modalities, such as functional MRI, dual-energy CT, perfusion CT and radiomics, it is possible to improve characterization of thymic epithelial tumors and other mediastinal tumors, assessment of tumor invasion into adjacent structures and detection of secondary lymph nodes and metastases. This review aims to illustrate the actual state of the art in diagnostic imaging of thymic lesions, describing imaging findings of thymoma and differential diagnosis.

## 1. Introduction

Thymoma is the most common primary neoplasm of anterior mediastinum, while thymic cyst, thymic hyperplasia, lymphoma and germ cells tumors account for the majority of the rest [1]. In 2015, the World Health Organization (WHO) classified thymomas into different types (A, AB, B1, B2 and B3) based on the morphology of epithelial tumor cells and the lymphocyte-to-tumor cells ratio [2], whereas the Masaoka staging system is based on the anatomic extent of tumor and invasion of the other organs. Both of the two systems are independent prognostic factors and are crucial when choosing the optimal treatment plan [3]. In fact, low risk (type A, AB and B1) and early stage (stage I and II) thymomas are usually treated with surgery alone, whereas type B2 and B3 and advanced stages (stages III and IV) require a multimodal approach [4]. Diagnostic imaging is essential for characterizing and staging thymic tumors, though many lesions cannot be defined by using information extrapolated from traditional CT and MR images. In the era of advanced imaging modalities (functional MRI, Dual-energy CT, perfusion CT, radiomics), it is possible to improve characterization of thymic epithelial tumors and other mediastinal tumors, assessment of invasion into adjacent structures and detection of lymph nodes and metastasis.

This review aims to illustrate the actual state of the art in diagnostic imaging of thymic lesions, describing imaging findings of thymoma and differential diagnosis. Literature research was done on PubMed by using the following key words: thymic tumor, dual-energy CT, spectral CT, perfusion CT, radiomics, MR. Articles were selected considering the maximum sample size and the most exhaustive reviews.

## 2. CT

Chest CT is the first imaging modality performed for a clinical or X-ray suspicion of a thymic lesion, thanks to its low-cost, wide availability and fast execution. The exam should include a scan without contrast medium and a post-contrast scan in late arterial phase, from diaphragmatic pillars to the base of the neck, in order to evaluate the presence of ectopic lesions and pleural, diaphragmatic metastases [5,6,7,8,9,10] (Figure 1).

Some authors tried to correlate CT morphological features with clinical information and Masaoka–Koga and WHO classifications. Han X. et al. analyzed 159 patients with thymomas who underwent preoperative CT and found that tumor size and pleural effusion were associated with patient’s symptoms; moreover, high risk (B2 and B3) and stage III and IV thymomas were larger, with calcifications and irregular contours, and showed infiltration of vessels and mediastinal fat if compared with low-risk (A, AB, B1) and stage I and II lesions. Contours, abutment ≥ 50% of mediastinal structures and adjacent lung abnormalities were risk factors for metastases and recurrence [11]. Kato T. et al. investigated 53 patients with Masaoka stage I to III thymoma and 5 patients with stage IVa with 6 cases of pleural recurrence within 76 months of follow-up. Tumor diameter and contact length between the tumor and the lung were measured, and they found that the contact was significantly longer in the group with pleural recurrence, whereas tumor diameter was not significantly different between recurrence and non-recurrence group [12]. Shen Y. et al. investigated the possibility of predicting stage III thymoma with CT, analyzing 66 patients, and they found that for pleural and pericardial invasion, an absence of space between the tumor and pleura/pericardium with pleural/pericardial effusion and thickening showed a specificity of 100%. A multilobular tumor convex to the lung with adjacent abnormalities was highly specific for lung invasion, whereas for vessel invasion, specificity was 100% for tumors abutting ≥ 50% of the vessel circumference [13]. Regarding lung invasion, Green D.B. et al. enrolled 54 patients with thymoma, evaluating the interface between tumor and adjacent lung parenchyma. According to Shen Y. et al., they found that multilobulated thymomas were more likely to invade the lung; however, specificity increased significantly using the criterion of multilobulation with at least one acute angle among them [14] (Figure 2). The capability to predict lung invasion on preoperative imaging is pivotal for surgical planning because lung invasion may require a wedge resection and diaphragmatic plication due to phrenic nerve involvement [15]. In the same way, the evaluation of vascular infiltration is pivotal when establishing a surgical plan and assessing its feasibility. Direct signs of vascular involvement include vascular encasement, endoluminal soft tissue and irregular lumen contour [16].

## 3. Dual-Energy CT

Dual-energy CT (DECT) indicates the acquisition of CT data derived from two different photon energies so that different materials can be better characterized by evaluating the difference in attenuation at these two energies [17]. Thanks to technical improvement, DECT images can be obtained with a radiation exposure not significantly different from that of a conventional single-energy CT. The scans can be performed by using different technical approaches: dual-source, single source with rapid switching, single source twin beam and dual-layer detector. Raw data can be elaborated at a dedicated workstation, thus achieving a set of monoenergetic images ranging from 40 to 140 keV and material-specific images, such as iodine and water-related. Given the increased attenuation of iodine at a low energy level, low energy series enhance lesion conspicuity with several advantages in oncologic imaging. Moreover, DECT is not affected by beam hardening artifacts, and it is possible to accurately quantify iodine uptake of tissue with better characterization of different lesions [18,19] (Figure 3).

Concerning our topic of interest, a few studies investigated the potential of DECT for characterizing thymic lesions.

Yen W. et al. investigated the efficacy of iodine quantification for differentiating thymoma, thymic carcinoma and thymic lymphoma. The authors enrolled 57 patients (16 low-risk thymomas, 15 high-risk thymomas, 14 thymic carcinomas and 12 lymphomas). All patients underwent dual-phase post-contrast DECT scans in arterial and venous phases, and the following parameters were analyzed: iodine-related Hounsfield Unit (IHU), iodine concentration (IC), mixed HU (MHU), slope HU curve and virtual non contrast (VNC) values. The results demonstrated that IHU, IC and MHU were significantly higher in patients with low-risk than high-risk thymoma, thymic carcinoma and lymphoma: IHU in the venous phase showed the best performance for differentiating low-risk thymomas from the other tumors, with an AUC of 0.893 and a cut-off of 34.3 HU, while IC in the venous phase performed well for differentiating low-risk thymoma from lymphoma, with an AUC of 0.969 and a cut-off of 1.25 mg/mL. The higher values of DECT parameters in low-risk thymomas can be explained by the high prevalence of short-spindled variant among types A and AB thymomas, which are composed by short spindle cells arranged in a hemangiopericytic structure with a rich vascularization [20].

Xie Y. et al. [21] investigated the possibility of DECT to distinguish thymoma from mediastinal lymphoma because there is a significant overlap between the two diseases on conventional single-energy contrast CT. A total of 39 patients (24 with thymoma and 15 with lymphoma) were enrolled. DECT scans of the thorax were performed both in post-contrast arterial and venous phases by using fast tube voltage switching between 80 and 140 kVp. The following parameters were assessed for each lesion: normalized IC (NIC), calculated as the ratio between IC of the lesion and aorta, slope of spectral curve between 40 and 140 keV and CT attenuation difference, which is the difference in attenuation between arterial and venous phase at 70 keV. Data analysis showed a significant difference in NIC between patients with thymoma and lymphoma in venous phase (0.49 ± 0.15 mg/mL vs. 0.28 ± 0.08 mg/mL, respectively) and a significant higher slope of the spectrum curve in patients with thymoma in arterial phase. The higher IC of thymoma in the venous phase can be explained by the coexistence of neoplastic cells and fibrous bands within thymic lobules, which delay the washout of contrast medium. On the contrary, lymphomatous tissue lacks fibrous septa, and therefore the lesion washes out quickly after arterial phase.

## 4. CT Perfusion

CT perfusion is a technique that measures the changes in tissue density after intravenous injection of contrast medium through a series of CT images repeated in the same body section [22,23,24,25]. The coverage along the *z*-axis is limited by the length of detector rows, from 2 cm to 16 cm with a 320-detector scanner. Most vendors have implemented two different protocols by using helical scans or axial “shuttle mode”, and in this latter case, the detector row moves back and forth, covering a larger anatomical area [26]. The enhancement of tissue depends on iodine uptake, which reflects its vascularization [27] and can be divided in two phases on the basis of contrast medium (CM) distribution in the intravascular and extravascular compartment [28]. In the first 40–60 s from the arrival of contrast medium, the enhancement is mainly attributable to the distribution of CM within intravascular space, whereas in the second phase, it is influenced by vascular permeability to CM [29]. The main quantitative parameters that can be extracted are blood flow (BF) and blood volume (BV), which are related to the first phase of tissue enhancement, whereas permeability surface (PS) and mean transit time (MTT) are related to the second phase. CT perfusion has wide applications in oncology, concerning lesion characterization, prognosis and response to treatment.

In this regard, Bakan S. et al. [1] enrolled 25 patients with a mass of the anterior mediastinum, observed in a previous CT, MR or FDG-PET exam. CT perfusion was performed by using a 128-row scanner with a coverage of 6.9–9.6 cm along the *z*-axis, depending on the lesion size. Final diagnoses after surgery/biopsy included seven thymomas, eight thymic hyperplasias, four lymphomas, three thymic carcinomas and three lung cancers. Concerning perfusion parameters, BV was significantly higher in thymoma than in lymphoma and BF and BV were significantly higher in thymoma than in all other malignancies. No significant difference was found between hyperplasia and thymoma. The lower values of the parameters investigated in all the tumors different from thymoma is probably due to a predominance of a necrotic and cystic component in the first group.

Yu C. et al. [30] performed both DECT and perfusion scans in 51 patients with different WHO types of thymoma and found that perfusion and spectral parameters of type A and AB thymoma were significantly higher than those of other subtypes, given their histopathological composition, as demonstrated by previous works. Moreover, PS of high risk thymoma was lower than that of thymic carcinoma because vessels inside thymic carcinoma are more immature and therefore much more fenestrated.

## 5. PET/CT

PET/CT in last years has gained importance in the diagnosis of thymic malignancies, integrating metabolic information of PET with anatomical details of CT. Contrast medium is not routinely administered for the CT part of PET/CT because contrast enhancement may introduce over correction artefacts, reducing the reliability of SUV [31]. The ^18^F-FDG tracer is the most used in clinical practice. Additionally, thymic carcinoids are ^18^F-FDG avid; however, in cases of later diagnosis, ^68^Ga-DOTATE is more sensitive for detection of metastases and for selection of patients susceptible to peptide receptor radionuclide therapy [32].

MFK Benveniste et al. [16] analyzed the ^18^F-FDG-PET/CT scans of 51 patients with thymic epithelial neoplasms (37 thymomas, 12 carcinomas and 2 carcinoids), correlating SUV with Masaoka stage and WHO classification. The analysis showed that higher focal FDG uptake was seen in B3 thymoma than in the other subtypes. Moreover, FDG uptake was higher in patients with carcinoma or carcinoids than in patients with thymoma. However, no significant correlation was found between focal FDG uptake and disease stage.

Similarly, Sung Y.M. et al. [33] analyzed 33 patients with thymic epithelial tumors (8 low-risk thymomas, 8 high-risk thymomas and 16 thymic carcinomas) who underwent ^18^F-FDG-PET/CT. The authors found that maximum SUVs of high- and low-risk thymomas were significantly lower than those of thymic carcinomas. Moreover, homogeneous ^18^F-FDG uptake within tumors was more frequently seen in thymic carcinomas.

Luzzi L. et al. [34] investigated 19 patients who underwent ^18^F-FDG-PET/CT for a lesion of the anterior mediastinum (6 low-risk thymomas, 7 high-risk thymomas, 3 lymphomas and 3 other primitive thymic tumors). The mean SUV of low-risk thymomas was significantly lower than high-risk thymomas and lymphomas (3.3 ± 0.5 vs. 13.5 ± 7 vs. 12.4 ± 4 respectively).

Moreover, a significant correlation was found between SUV and Masaoka stages. The authors concluded that an anterior thymic mass with SUV < 5 is associated with low-risk thymomas and Masaoka stage I-II and therefore susceptible to upfront surgery. On the contrary, for lesions with SUV > 5 and an infiltrative aspect on CT, open biopsy is mandatory for a differential diagnosis between lymphoma and high-risk thymoma.

## 6. MRI

MRI is not the prime exam performed to detect a mediastinal lesion, except in case of renal failure and iodine allergy [35]. However, MRI, thanks to the superior soft-tissue contrast resolution, may give an advantage, if compared with CT, in distinguishing between solid and cystic thymic lesions and in evaluating tumor capsule and infiltration of vessels, pleura and pericardium [36,37,38]. Moreover, with the presence of diffuse thymic enlargement with convex borders, MRI can distinguish between thymoma and hyperplasia using dual-echo chemical shift sequences for identifying microscopic fat inside the gland, which is present only in case of hyperplasia; conversely, this diagnosis is almost impossible on CT [39].

Thymomas appear as focal lesion with intermediate signal intensity on T1-weighted sequences and with high signal intensity on T2-weighted sequences, eventually with necrosis, hemorrhage and cystic areas.

Some authors try to distinguish benign and malignant mediastinal neoplasms by using DWI and ADC values with different thresholds; however, there is a significant overlap between different cancers and also between low-grade, high-grade thymomas and thymic carcinomas [40,41,42].

DCE-MRI can be used to differentiate anterior mediastinal masses analyzing time-signal intensity curves (TICs). Yabuuchi H. et al. [43] analyzed 48 anterior mediastinal masses (23 low-risk thymomas, 9 high-risk thymomas, 6 thymic carcinomas, 7 lymphomas and 3 malignant germ cell tumors) comparing DCE-MRI, DWI and FDG-PET/CT. For each lesion, the following parameters were investigated: maximum diameter, presence of capsula/septa on T2-weighted images, TICs, ADC and SUVmax. No significant difference was found in ADC values either in the presence or absence of capsules between thymic tumors, lymphomas and germ cell tumors. The wash-out pattern on DCE was seen only in thymic epithelial tumors. SUVmax of thymomas was significantly lower than that of the other tumors (best cut-off: 11.6); moreover, the maximum diameter of all thymic tumors was significantly lower than that of lymphoma and germ cell tumors (best cut-off: 6.8 cm). Considering TICs, SUV max and maximum diameter, the accuracy for distinguishing different tumor histotypes reaches about 90%.

Sakai S. et al. [44] investigated the potential role of DCE-MRI for differential diagnosis of anterior mediastinal tumors in 59 patients with 31 thymomas, 14 thymic carcinomas, 7 lymphomas, 4 germ cell tumors and 3 thymic carcinoids. Post-contrast sequences were performed in the axial plane at 30 s interval for 5 min. Similar to Yabuuchi H. et al., these authors found that thymomas had an early enhancement peak with a following wash-out. All the other lesions, including thymic carcinomas and carcinoids, had a later peak (1.5 ± 0.9 min for thymomas vs. 3.2 ± 1.2 min for other lesions) (Figure 4).

However, no significant difference was found between different histological subtypes of thymoma.

## 7. Radiomics

Radiomics aims to identify subtle differences in radiological images that cannot be perceived by human eyes. In oncology, there may be a link between pixels relationship and biological characteristics of tumors. This field is constantly growing thanks to technological development and is very much promises to provide information about tumor diagnosis, prognosis and response to treatment [45,46,47].

Some recent research investigated the possibility of this technique for diagnosing thymic lesions to overcome limitations of qualitative interpretation of CT and MRI studies.

Iannarelli A. et al. [48] analyzed the relationship between texture analysis (TA) parameters applied to CT and WHO and Masaoka classifications. A total of 12 patients were affected by thymoma and 4 by thymic carcinomas. Thymic tumors were manually contoured on axial contrast-enhanced CT in venous phase, and the following TA parameters were evaluated: mean, standard deviation (SD), kurtosis, skewness, entropy, MPP. Results showed that WHO staging was correlated with mean, standard deviation and MPP, whereas Masaoka classification was correlated with mean, MPP, skewness and kurtosis.

Wang X et al. [49] analyzed 199 patients with thymoma, and TA parameters were extracted by non-enhanced CT scans and post-contrast ones with a delay of 30 sec from contrast medium injection. High-risk thymomas were found to be more heterogeneous than low-risk thymomas, and both TA parameters of non-contrast and post-contrast scans proved to effective for discriminating between the two groups (AUC = 0.801 for non-enhanced scan and 0.827 for enhanced scans).

Xiao G. et al. [50] enrolled 189 patients with thymic epithelial tumors (73 low-risk thymomas, 69 high-risk thymomas and 47 thymic carcinomas) who underwent pre-operative MRI. Lesions were segmented manually on axial T2 weighted and T2 weighted fat-suppressed sequences. Feature selection was performed using machine learning software. Two models based on 125 and 69 features were constructed for predicting pathological classification and TNM staging, respectively.

The models achieved an AUC of 0.880 and 0.771 for distinguishing among low-risk and high-risk thymomas, thymic carcinomas and early from advanced stage disease.

## 8. Conclusions

New imaging techniques have significantly improved the possibilities to stage thymic lesions in the preoperative phase, allowing better therapeutic management of these patients. Radiomics applied to CT and MRI images is very promising, although the standardization of software is required for its application in clinical practice.

## Figures and Tables

**Figure 1 cancers-13-03599-f001:**
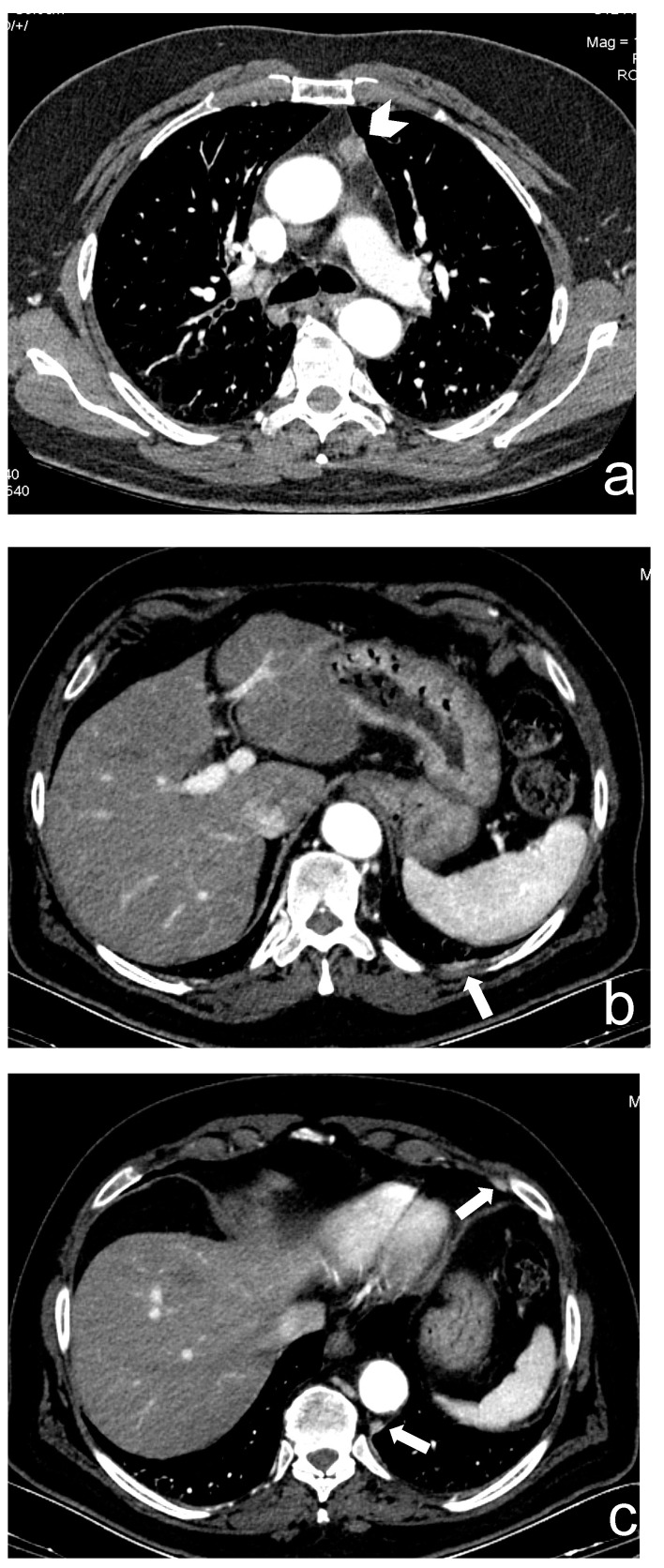
(**a**–**c**) A 66-year-old man who underwent robotic surgery in March 2016 for a thymic basaloid carcinoma (stage I, arrowhead in (**a**)). In June 2020, small bilateral pleural metastatic lesions appeared without pleural effusion (arrows in (**b**,**c**)).

**Figure 2 cancers-13-03599-f002:**
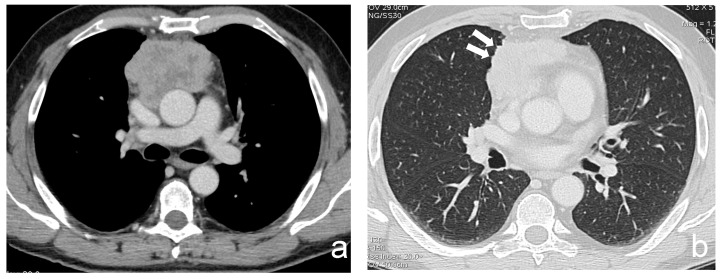
(**a**,**b**) A 63-year-old man with a poor differentiated squamous thymic carcinoma invading the pericardium and the lung (stage IIIa). Margins of the lesion are multilobulated and indented at the interface with the lung; the latter CT finding is better demonstrated using a lung window (arrows).

**Figure 3 cancers-13-03599-f003:**
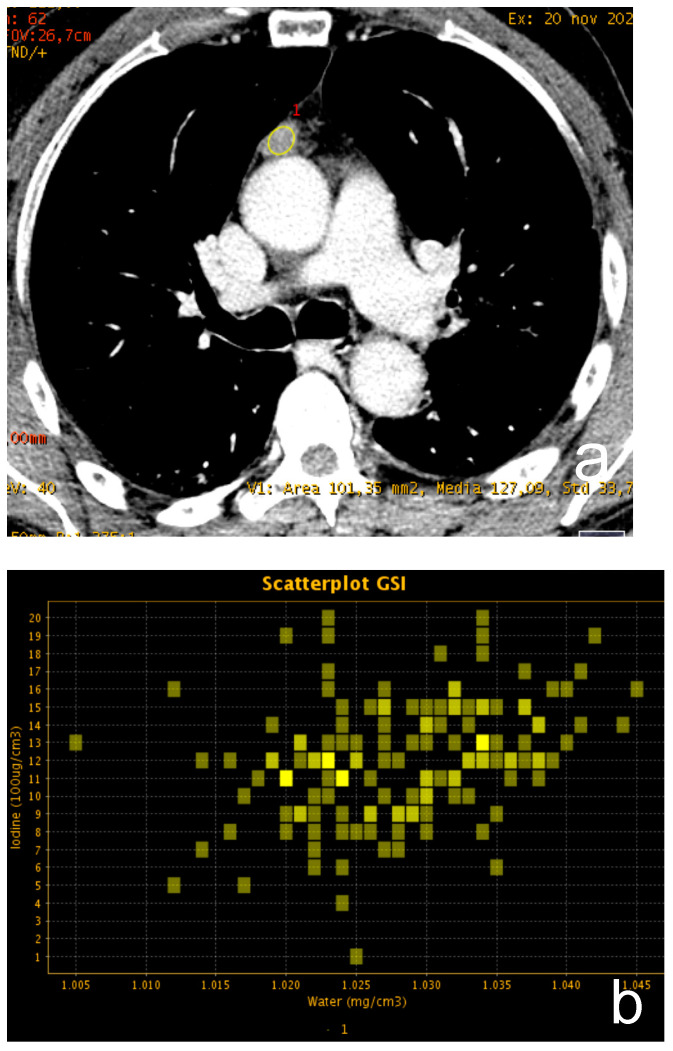
(**a**–**c**) A 62-year-old man with a small thymic lesion with convex margins. Dual-energy post-contrast images (scatterplot in (**a**) and spectral curve in (**b**) shows a mild enhancement of the lesion (mean iodine concentration 1.2 mg/dL). Histopathology revealed a medullary type A thymoma with capsule infiltration (Masaoka stage IIa).

**Figure 4 cancers-13-03599-f004:**
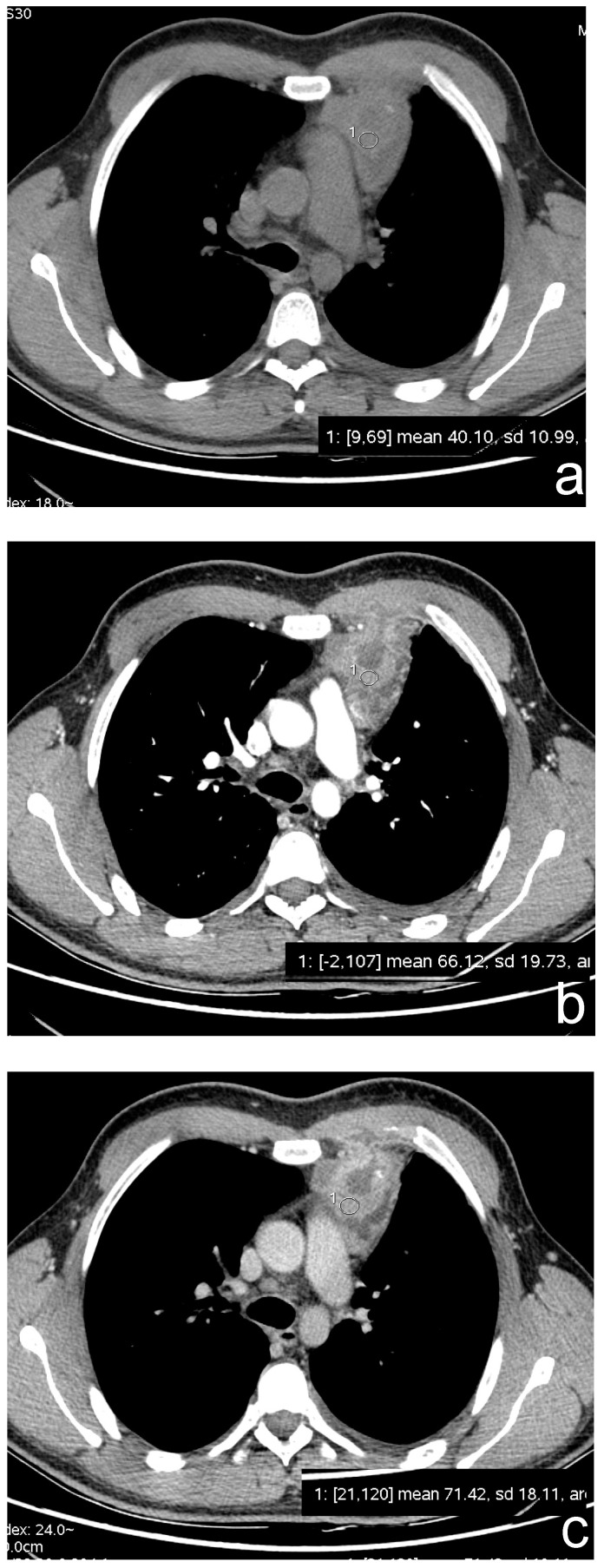
(**a**–**c**) A 23-year-old man with a partially cystic lesion of the anterior mediastinum. CT scans without contrast (**a**) and after contrast medium administration in arterial (**b**) and delayed (3 min, (**c**)) phases show late enhancement peak of the solid component (ROI 1). Surgery revealed a seminoma.

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
