# Peer review of "Advancement in Diagnostic Imaging of Thymic Tumors"

_cancers, 2021, doi:10.3390/cancers13143599_

Round 1

Reviewer 1 Report

In this manuscript the Authors provide a through review on the state of the art of diagnostic imaging of thymic tumors. It could be of interest to improve the text by adding some information concerning the role of nuclear medicine in the imaging of thymic tumors.

Author Response

The authors thank the reviewer for this comment. A paragraph about the role of 18FDG-PET-CT in diagnosing thymic tumors has been added to the text.

Reviewer 2 Report

This manuscript submitted by Gentili et al, discussed the different diagnostic imaging techniques to characterize and stage thymic tumors. The review article has explained and illustrated every aspect of the topic in detail. The manuscript provides some interesting information in clinical field, hence, manuscript may accept in present form.

Author Response

Authors thank the reviewer for this comment.

Reviewer 3 Report

The article "Advancement in diagnostic imaging of thymic tumors" aims to illustrate the state of the art of imaging of thymus tumors, also providing the data available so far on the differential diagnosis between the different histological entities. The work is well written and comprehensive for all traditional and non-traditional imaging techniques for the diagnosis and characterization of such tumors. However, the role of PET with 18F-FDG or using alternative radiopharmaceuticals (eg 68Ga-DOATATOC / DOTATATE) is not mentioned at all. Given the large literature available on this topic, I believe that a new paragraph should be added in the text explaining the role of metabolic imaging in the characterization of thymic neoplasms, possibly illustrating the advantages and disadvantages of positron emission tomography over traditional imaging.

Author Response

Authors thank the reviewer for this comment. A paragraph about the role of PET/CT for diagnosing thymic neoplasms has been added to text explaining the potential advantages over traditional imaging.